# CELL-E 2: Translating Proteins to Pictures and Back with a Bidirectional Text-to-Image Transformer

**Emaad Khwaja**[*‡]
emaad@berkeley.edu

**Yun S. Song**[†‡]
yss@berkeley.edu

**Aaron Agarunov**[§]
agarunov.aaron@gmail.com

**Bo Huang** [¶‖**]
bo.huang@ucsf.edu

## Abstract

We present CELL-E 2, a novel bidirectional transformer that can generate images depicting protein subcellular localization from the amino acid sequences (and *vice versa*). Protein localization is a challenging problem that requires integrating sequence and image information, which most existing methods ignore. CELL-E 2 extends the work of CELL-E, not only capturing the spatial complexity of protein localization and produce probability estimates of localization atop a nucleus image, but also being able to generate sequences from images, enabling *de novo* protein design. We train and finetune CELL-E 2 on two large-scale datasets of human proteins. We also demonstrate how to use CELL-E 2 to create hundreds of novel nuclear localization signals (NLS). Results and interactive demos are featured at https://bohuanglab.github.io/CELL-E_2/.

## 1 Introduction

Subcelllular protein localization is a vital aspect of molecular biology as it helps in understanding the functioning of cells and organisms [1]. The correct localization of a protein is critical for its proper functioning, and mislocalization can lead to various diseases [2]. Protein localization prediction models have typically relied on protein sequence data [3, 4] or fluorescent microscopy images [5, 6] as input to predict which subcelleular organelles a protein would localize to, designated as discrete class labels [7, 8]. The CELL-E model was markedly different in that it utilized an autoregressive text-to-image framework to predict subcellular localization as an images [9], thereby overcoming bias from discrete class labels derived from manual annotation [10]. Furthermore, CELL-E was capable of producing a 2D probability density function as an image based on localization data seen throughout the dataset, yielding more a far more interpretable output for the end user.

Although novel, CELL-E was inherently restricted by the following limitations:

**Autoregressive Generation.** Alongside other autoregressive models [11–14], CELL-E was limited by slow generation times and unidirectionality. When provided with input, CELL-E required a separate step for each image patch (256 for the output image composed of tokens of size $16 \times 16$).

---

[*]UC Berkeley - UCSF Joint Bioengineering Graduate Program

[†]Department of Statistics, UC Berkeley, CA 94720

[‡]Computer Science Division, UC Berkeley, CA 94720

[§]Department of Pathology, Memorial Sloan Kettering Cancer Center, 10065

[¶]Department of Pharmaceutical Chemistry, UCSF, San Francisco, CA 94143

[‖]Department of Biochemistry and Biophysics, UCSF, San Francisco, CA 94143

[**]Chan Zuckerberg Biohub - San Francisco, San Francisco, CA 94158

37th Conference on Neural Information Processing Systems (NeurIPS 2023).

This slow image generation severely limits the ability of CELL-E to perform a high-throughput mutagenesis screening.

**Unidirectional Prediction.** The unidirectional nature of CELL-E allowed for predictions to be made in response to an amino acid sequence, however it may be of interest to biologists to make predictions of sequence given a localization pattern. Such capability would be advantageous for those working in a protein engineering domain [15, 16]. One could imagine a researcher wanting to know the optimal localization sequence to append to a protein on either the N or C terminus [17] while maintaining essential function within an active site region, as well as reducing the chance of off-target trafficking.

**Limited Dataset.** CELL-E utilized the OpenCell dataset [18], a relatively small dataset. Vision transformers often require large amounts of data to make robust predictions [19], however a small dataset was utilized in the original model. This led to a degree of overfitting and prediction bias based on the limited diversity in localization patterns of the original dataset.

**Present Work.** As in CELL-E, our method CELL-E 2 is able to generate accurate protein localization image prediction as illustrated in Fig. 1, but it differs from CELL-E by employing a non-autoregressive (NAR) paradigm which improves the speed of generation. Similar to CELL-E, we retrieve embeddings from a pre-trained protein language model and concatenate these with learned embeddings corresponding to image patch indices coming from a nucleus (a subcellular organelle containing DNA [20]) image and protein threshold image encoders (Fig. 2). We then apply masking to both the amino acid sequence as well as the threshold image in an unsupervised fashion, and reconstructed tokens are predicted in parallel, allowing for generation in fewer steps. This also allows for bidirectional prediction, (sequence to protein threshold image or protein threshold image to sequence). Additionally, to improve the predictive performance we utilize a larger corpus of data, the Human Protein Atlas (HPA) [21] in pre-training to expose the model to a higher degree of localization diversity, and finetune on the OpenCell dataset [18], which better represents natural protein localization because it is acquired from live instead of fixed cells. We explore multiple strategies towards finetuning which serves to generally inform task-specific refinement text-to-image models in Section 5.3.

## 2 Related Work

### 2.1 Protein Language Models

Embeddings from unsupervised protein language models can be used to predict and analyze the properties of proteins, such as their structure, function, and interactions [22]. By exploring the hidden patterns and relationships within these sequences, protein language models can help to advance our understanding of the complex world of proteins and their roles in various biological processes. Masked language modelling has been particularly successful. Ankh [23], ProtT5 [24], ProGen [25], ESM-2 [26], and OmegaFold [27] are examples of recent models which use masked langauge approaches. Embeddings from ESM-2 and Omegafold in particular have been used for structural prediction, indicating hierarchies of information beyond the primary sequence contained in the embeddings [28].

### 2.2 Protein Localization Prediction

Protein localization prediction via machine learning is an emerging field that uses computational algorithms and statistical models to predict the subcellular spatial distribution of proteins [29]. This is an essential task in biology, as the subcellular localization of a protein plays a crucial role in determining its function and interactions with other proteins [30, 31] The prediction of protein localization is performed by analyzing protein sequences, amino acid composition, and other features that can provide clues about their subcellular location. Machine learning algorithms are trained on large datasets of labeled proteins to recognize patterns and make predictions about the subcellular location of a protein. This field has the potential to improve our understanding of cellular processes, drug discovery, and disease diagnosis.

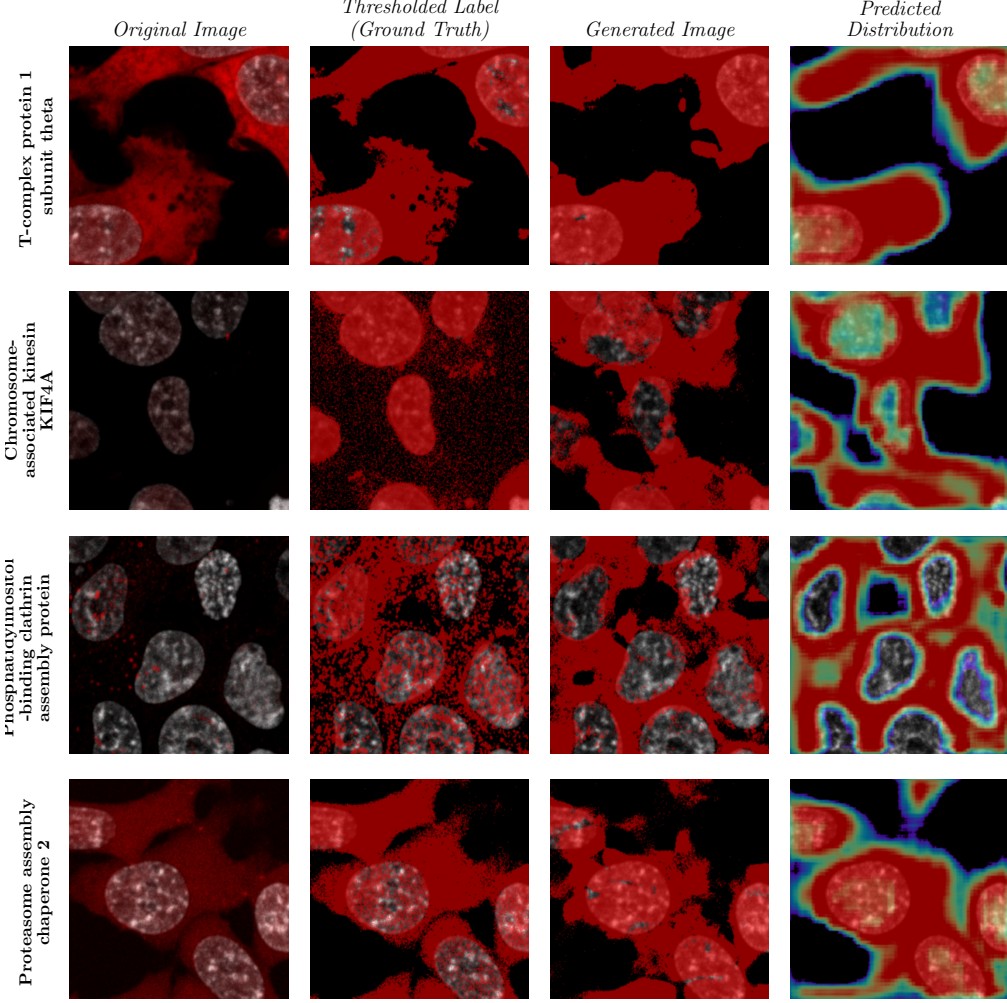

Figure 1: Localization predictions from CELL-E 2 (`HPA Finetuned (Finetuned HPA VQGAN)_480`) on randomly chosen validation set proteins from the OpenCell dataset. All images feature the Hoescht-stained nucleus image as a base. The "Original Image" column shows the fluorescently labelled protein from the dataset. The "Thresholded Label" shows pixels greater than the median value. This serves as the ground truth for the model during training. "Generated Image" is the image specifically predicted by CELL-E 2 and is compared against the thresholded ground truth image. "Predicted Distribution" is the latent space interpolation of the binary threshold image tokens which uses which utilizes the output logits of CELL-E 2. See Fig. S1 for colorbars corresponding to all plots in this work.

Recently, attention-based methods have demonstrated the ability to predict localization from a sequence [32], enabling the use of long context information when compared to convolutional-based counterparts [33–35]. These methods, however, predict localization as discrete classes rather than as an image. CELL-E, on the contrary, does not utilize existing annotation and provides a heatmap of the expected spatial distribution on a per-pixel basis [9]. This approach enables learning at scale by eliminating the bottleneck of manual annotation while also circumventing label bias.

## 2.3 Text-to-Image Synthesis

Recent gains in the text-to-image field have have largely been made by autoregressive models [11, 13], which correlate text embeddings with image patch embeddings, as well as diffusion models, [14, 36–39], which condition on sentence embeddings to gradually synthesize images from random noise.

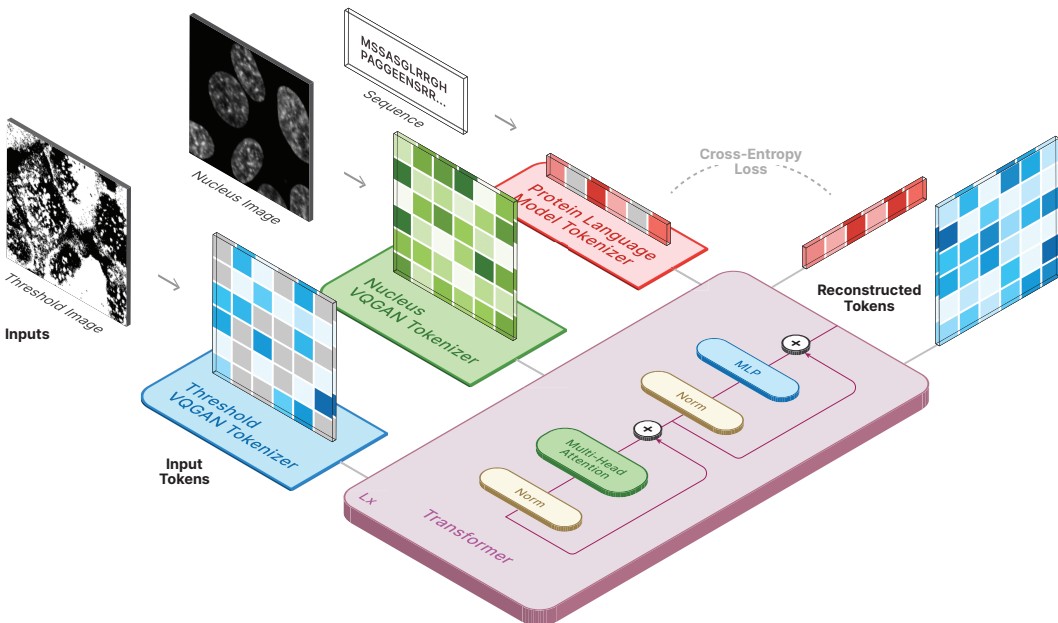

Figure 2: Depiction of training paradigm for CELL-E 2. Gray squares indicate masked tokens. Loss is only calculated on masked tokens in the sequence and protein threshold image.

Other works implement non-autoregressive models (NAR), which take advantage of a masked reconstruction procedure, similar to BERT [40], where the model is tasked with predicted randomly masked portions of an input image. These types of models are particulalry advantageous because they enable parallel decoding, allowing images to be synthesized in relatively view steps when compared to autoregressive models. Furthermore, NAR models are not bound to a particular direction of synthesis like autoregressive models, which only perform next-token prediction. CogView2 [41] utilizes a modified transformer architecture where attention on masked tokens is eliminated. MUSE [42] builds on MaskGIT [43] by concatenating a pre-trained text embedding to a token masked representation of a corresponding image. It uses a vanilla transformer architecture [44] and yielded state-of-the-art image synthesis performance in terms of FID and human evaluation.

## 3 Datasets

We pretrained our model on protein images from the Human Protein Atlas (HPA) [45], which covers various cell types and imaging conditions using immunofluorescence microscopy1. We then finetuned on the OpenCell dataset [18], which has a consistent modality using live-cell confocal microscopy of endogenously tagged proteins. To ensure generalization to new data, we followed the homology partitioning method of Almagro Armenteros et al. [35]. We used PSI-CD-HIT [46] to cluster HPA proteins at ($\geq 50\%$) sequence similarity and randomly split the clusters into 80/20 train/validation sets. We applied the same clustering and splitting to the OpenCell proteins, matching the train/validation labels from HPA. For proteins present in OpenCell but not HPA ($= 176$), we assigned the protein based on the other labels in the cluster. Any remaining unassigned proteins ($= 1$) were assigned to the training set. See Appendix A for more details about the datasets and preprocessing.

## 4 Methods

CELL-E 2 (Fig. 2) is a masked NAR transformer model, which upgrades the capabilities of CELL-E, an autoregressive *decoder-only* model [47]. Due to the NAR nature of the model, CELL-E 2 is capable of both image generation (sequence to image), as well as sequence prediction (image to sequence).

### 4.1 Amino Acid Sequence Embeddings

Proteins are biological molecules which are comprised of individual amino acids. CELL-E 2 utilizes embeddings from ESM-2 [26], where amino acid molecules are denoted with individual letter codes (e.g. A for alanine) [48]. We opt to use frozen amino acid embeddings for the prediction task, which has been demonstrated to yield superior reconstruction performance in text-to-image models [9, 37, 42]. The embeddings obtained from a protein language model contain valuable information about amino acid residues, biochemical interactions, structural features, positional arrangements, as well as other characteristics like size and complexity [22]. We train models of varying size based on the released ESM-2 checkpoints (See Section 5). The output of the final embedding layer per respective model is used as the amino acid sequence embedding.

### 4.2 Image Tokenization

Just as in Khwaja et al. [9], we utilize a nucleus image, which serves as a spatial reference with which a binarized protein threshold image is associated. We chose this in order to parallel the type of images which are typicall acquired in a wet lab scenario.

We also utilize VQGAN autoencoders [49] trained on both the HPA and OpenCell datasets, respectively. VQGAN surpasses other quantized autoencoders by incorporating a learned discriminator derived from GAN architectures [50]. Specifically, the Nucleus Image Encoder employs VQGAN to represent $256 \times 256$ nucleus reference images as $16 \times 16$ image patches, with a codebook size of ($n = 512$) image patches. To enable transfer learning, we explore finetuning strategies these VQGANs in Section 4.5.

The protein threshold image encoder acquires a compressed representation of a discrete probability density function (PDF) that maps per-pixel protein positions, presented as an image. We binarize the image based on the median pixel value of the image (see Appendix A.4). We utilize a VQGAN architecture identical to the Nucleus VQGAN to estimate the entire set of binarized image patches to denote local protein distributions. These VQGANs are trained until convergence, and the discrete codebook indices are used for the CELL-E 2 transformer. Hyperparameters (Table S1, Table S2, Table S3) and training details can be found in Appendix B.2.

### 4.3 Input Masking Strategy

We adopt a cosine-scheduling technique for masking image tokens, which has been used by other works. The probability of an image patch being masked is determined by a cosine function, favoring high masking rates with an expected masking rate of $64\%$ [42, 43]. This technique provides various levels of masking during the training process, exposing the model to spatial context for masked language tokens.

Of similar interest as image prediction, sequence in-filling with respect to a localization pattern has potential to significantly augemnt a biologist's workflow. Typically, protein localization sequences are found through sequence similarity searches with proteins that have known localization in particular organelles [51–53] or via experimentation [54, 55]. CELL-E 2's bidirectionality enables the model to make predictions for image with respect to amino acid sequences, as well as sequence predictions conditioned on images. This enables an entirely new paradigm in protein engineering which relies on image information. To achieve this, we also mask the language tokens along with the protein threshold image tokens. We experimented with using the same cosine function for image masking but found it led to numerical instability and vanishing gradients. Therefore, we linearly scaled the cosine function to ensure the maximum masking rate matched $15\%$ masking rate used to train ESM-2.

### 4.4 Base Transformer

The base transformer is an NAR model in which the embedding dimension is set to the embedding size of the pretrained language model used. We utilized two types of masking tokens. For masking the amino acid sequence, we leveraged the mask token which already exists within the ESM-2 dictionary, designated as `<MASK_SEQ>`. The VQGAN does not contain a masking token within its codebook, so to represent it, we add an additional entry in the image token embedding space of CELL-E 2 (with $n + 1$: $(512 + 1 = 513$), where $n$ is the number of tokens in the VQGAN codebook),

and designate the final token as `<MASK_IM>`. We additionally create an embedding space of length 1 for the `<SEP>` token which is appended to the end of the amino acid sequence. Training details can be found in Appendix B.2.

We sample from this transformer by strategically masking positions in the image or sequence (see Appendix B.1). The logit values for the image prediction are used as weights for the threshold image patches to produce a predicted distribution (Fig. 1, Fig. S5).

### 4.5 Finetuning

We sought to leverage useful information from both HPA and OpenCell. HPA contains many proteins (17,268), but is potentially subject to inaccuracies, fundamentally because of the immunohistochemistry used for staining requires several rounds of fixation and washing [21]. This means the proteins are not observed in a live cell; are subject to signal loss, artifacts, and/or relocalization events; and therefore may not necessarily represent the true nature of protein expression and distribution within a cell [56]. The OpenCell dataset, while comparatively smaller, overcomes these issues by using a split-fluorescent protein fusion system allows for tagging endogenous genomic proteins, maintaining local genomic context, and the preservation of native expression regulation for live cell imaging [18, 57]. We therefore utilize the HPA dataset for pretraining, and then finetuned on OpenCell.

Optimally finetuning withinin the text-to-image domain remains an open question. The use of multiple models makes it difficult to pin down the correct strategy. Contemporary efforts utilize pretrained checkpoints to fine-tune on domain specific data [58–60]. Chambon et al. [61] reported improved synthesized image fidelity when fine-tuning the U-net of a text-to-image diffusion model, but similar fine-tuning strategies have not been explored for patch-based methods. We report our findings in Section 5.3.

## 5 Results

Similar to CELL-E, we cast the embedding spaces for the image tokens at the same size as the ones used by the pre-trained language model. The size of the embedding vectors ("Hidden Size") for each model was chosen based on the publicly available ESM-2 checkpoints. For instance, a CELL-E 2 model with hidden size $= 480$ uses `esm2_t12_35M_UR50D`, which corresponds to a 35M parameter model with 12 attention layers. Khwaja et al. [9] demonstrated a positive relationship between the number of attention layers (designated "Depth") in the base transformer and the image prediction performance. The maximum depth was set based on our available GPU memory capacity. We refer to models using the name format "`Training Set_Hidden Size`".

### 5.1 Protein Localization Image Prediction Accuracy

To predict the protein localization image, we provide CELL-E 2 with the protein sequence and nucleus image, and fill the protein image token positions with `<MASK_IM>` tokens (Fig. S3).

We evaluated the models on several image metrics (see Appendix C.1) that measure the quality and diversity of the generated protein images (Table 1). Additionally, we assessed the model's generalization capabilities by testing them on the other dataset (HPA-trained model on OpenCell and *vice versa*) (Table S4). We report the results for each model on its respective dataset. We observed a significant positive effect of depth on performance across all metrics and datasets. The models with hidden sizes of 480 and 640 achieved the highest scores, with no significant difference between them. However, on the HPA dataset, `HPA_640` surpassed the `HPA_480` model in more categories. On the OpenCell dataset, `OpenCell_480` performed better than the `OpenCell_640`.

We conducted a visual examination of the generated protein images as depicted in Figures Fig. S6 and Fig. S7. Among the models, OpenCell demonstrated a higher visual resemblance and consistency with its respective ground truth labels, although they exhibited low entropy in the predicted distribution. This indicates that while these models accurately identified the correct tokens with high probability, they struggled to account for the uncertainty and variety inherent in other valid choices, possibly due to a tendency for rapid overfitting which hindered their generalizability.

Table 1: Validation Set Image Prediction Accuracy. MAPE: mean absolute percentage error, MAE: mean absolute error, SSIM: structural similarity index measure, FID: Fréchet inception distance, IS: inception score.

| Dataset | Hidden Size | Depth | Nucleus Proportion MAPE | Image MAE | PDF MAE | SSIM | FID | IS |
|---|---|---|---|---|---|---|---|---|
| HPA | 480 | 68 | **0.0257 ± 0.0250** | 0.3340 ± 0.0788 | 0.2846 ± 0.0985 | 0.2633 ± 0.1781 | 12.0332 | 2.2900 ± 0.0410 |
| | 640 | 55 | 0.0294 ± 0.0278 | **0.3283 ± 0.0805** | **0.2842 ± 0.0991** | **0.2826 ± 0.1827** | 21.7942 | 2.2618 ± 0.0364 |
| | 1280 | 25 | 0.0370 ± 0.0360 | 0.3622 ± 0.0799 | 0.2967 ± 0.0985 | 0.2645 ± 0.1857 | **1.5161** | **2.5440 ± 0.0490** |
| | 2560 | 5 | 0.0818 ± 0.0794 | 0.3516 ± 0.0792 | 0.3104 ± 0.0904 | 0.2558 ± 0.1619 | 23.7977 | 2.1578 ± 0.0290 |
| OpenCell | 480 | 68 | 0.0161 ± 0.0148 | **0.4953 ± 0.0064** | **0.3620 ± 0.1168** | **0.1220 ± 0.1188** | 1.5844 | **2.6069 ± 0.1175** |
| | 640 | 55 | **0.0159 ± 0.0136** | 0.4995 ± 0.0006 | 0.3785 ± 0.1008 | 0.1011 ± 0.1012 | 2.6966 | 2.0974 ± 0.0981 |
| | 1280 | 25 | 0.0272 ± 0.0223 | 0.4996 ± 0.0010 | 0.4359 ± 0.0700 | 0.0694 ± 0.0472 | 8.9102 | 1.3712 ± 0.0432 |
| | 2560 | 5 | 0.0584 ± 0.0511 | 0.4996 ± 0.0005 | 0.4145 ± 0.0889 | 0.0890 ± 0.0667 | 9.5116 | 1.4176 ± 0.0329 |

Table 2: Validation Set Masked Sequence In-Filling

| Dataset | Hidden Size | Depth | Sequence MAE | Cosine Similarity |
|---|---|---|---|---|
| HPA | 480 | 68 | 0.8628 ± 0.0951 | 0.9504 ± 0.0237 |
| | 640 | 55 | 0.7917 ± 0.1245 | 0.9577 ± 0.0216 |
| | 1280 | 25 | 0.6512 ± 0.1794 | 0.9708 ± 0.0163 |
| | 2560 | 5 | **0.5759 ± 0.2322** | **0.9722 ± 0.0210** |
| OpenCell | 480 | 68 | 0.7507 ± 0.1709 | 0.9533 ± 0.0285 |
| | 640 | 55 | 0.6641 ± 0.1764 | 0.9610 ± 0.0272 |
| | 1280 | 25 | 0.5698 ± 0.2016 | 0.9709 ± 0.0220 |
| | 2560 | 5 | **0.4950 ± 0.2456** | **0.9711 ± 0.0271** |

We also observed models had stronger performance with respect to the dataset on which they were trained. Notably, the model trained on the HPA dataset outperformed the OpenCell-trained model on the OpenCell dataset, showcasing lower PDF MAE values across all categories. This HPA model also displayed lower FID on the OpenCell validation set, underscoring the advantage of having a more extensive dataset even under differing imaging conditions. The `OpenCell_480` model achieved the best scores in half of the evaluated metrics: MAPE, MAE, SSIM, and IS.

## 5.2 Masked Sequence In-Filling

To test each model's sequence learning, we used a masked in-filling task similar to the training task. Similar to Section 5.1, we provide CELL-E 2 with a randomly masked (15%) sequence, an unmasked nucleus image, and an unmasked protein threshold image. To select the sequence prediction, we perform a weighted random sampling operation from the 3 amino acids with the highest predicted probabilities. We measured the accuracy as the percentage of correct predictions (noted as "Sequence MAE", see Appendix C.2). We then embedded each reconstructed sequence with `esm2_t36_3B_UR50D`, the largest model we could fit in memory, with 3B parameters, 36 layers and an embedding dimension of 2560. We computed the mean cosine similarity between the embeddings of the original and reconstructed sequences at masked positions. We show validation results in (Table 2) and all results in (Table S6).

Most models had low performance on this task in terms of reconstruction. This is understandable because the models learned to generate amino acids that were common or frequent in the dataset, but not necessarily correct for the specific sequence. On the other hand, we observed values close to 1 for the cosine similarity, indicating that the predicted amino acids had similar embedding values to the original ones at the masked positions. This could be because the models learned to capture some semantic or structural features of the amino acids, such as polarity or charge, that were reflected in the embedding space and contributed to the biological function of the sequence. Models that used the embedding model with an embedding dimension of 2560 had the best performance. For example, `OpenCEll_2560` had the best performance on both metrics, with a MAE of .4950 and cosine similarity of .9711. When compared to randomly selected amino acids for each position (Table S7), we note significantly higher Sequence MAE and Cosine Similarity.

We also note that the reconstruction ability does not improve the performance of the original language models (Table S5). This may be a result of the combined image/sequence loss used during training or a smaller corpus of data compared to datasets used for the training the original language model. Evaluation results across both datasets can be found in (Table S6).

Table 3: OpenCell Validation Set Image Prediction Accuracy after Finetuning

| Fine-Tuned | Threshold Image Encoder | Nucleus Proportion MAPE | Image MAE | PDF MAE | SSIM | FID | IS |
|---|---|---|---|---|---|---|---|
| No | HPA | $0.0181 \pm 0.0168$ | $0.4154 \pm 0.0594$ | $0.3887 \pm 0.1270$ | $0.1250 \pm 0.1149$ | 3.9509 | $2.1739 \pm 0.1255$ |
| No | OpenCell | $0.0161 \pm 0.0148$ | $0.4953 \pm 0.0064$ | $0.3620 \pm 0.1168$ | $0.1220 \pm 0.1188$ | **1.5844** | $2.6069 \pm 0.1175$ |
| Yes | HPA | $0.0166 \pm 0.0151$ | $0.3776 \pm 0.0834$ | $\mathbf{0.3477 \pm 0.1268}$ | $0.1869 \pm 0.1503$ | 17.4075 | $2.9113 \pm 0.1199$ |
| Yes | OpenCell | $\mathbf{0.0159 \pm 0.0156}$ | $0.4996 \pm 0.0006$ | $0.3506 \pm 0.1208$ | $0.1574 \pm 0.1372$ | 2.5026 | $2.7168 \pm 0.1137$ |
| Yes | HPA Finetuned | $0.0170 \pm 0.0160$ | $\mathbf{0.3449 \pm 0.1305}$ | $0.3487 \pm 0.1340$ | $\mathbf{0.1881 \pm 0.1541}$ | 19.2683 | $\mathbf{3.6083 \pm 0.2013}$ |

## 5.3 Finetuning

While the HPA dataset contains information about a wide variety of proteins, the model does not innately perform as well on the OpenCell data. We considered the potential of utilizing an HPA-trained model and finetuning on the OpenCell data, thereby introducing a wider protein context than what is found in the OpenCell data alone while adapting to the imaging conditions and cell type found within the new dataset. We experimented with different finetuning strategies for CELL-E 2 on the OpenCell dataset. We used the pre-trained HPA checkpoint as the starting point for all finetuned models, continuing training on the OpenCell train set. We also evaluated the pre-trained HPA and OpenCell checkpoints without any finetuning as baselines. The finetuned models differed in how they updated the image encoders:

- `HPA Finetuned (HPA VQGAN)`: we kept the original VQGAN image encoders from the HPA checkpoint.

- `HPA Finetuned (OpenCell VQGAN)`: we replaced the image encoders with VQGANs trained only on OpenCell data.

- `HPA Finetuned (Finetuned HPA VQGAN)`: we finetuned the HPA image encoders while keeping the rest of the model frozen, then froze the image encoders and update the transformer weights.

Fig. S8 shows image predictions on an OpenCell validation protein for models with hidden size = 480. Surprisingly, the pretrained HPA model already achieved strong performance on the OpenCell dataset without any finetuning (see Table S8).

The best results were obtained by utilizing a pre-trained HPA checkpoint. We first finetuned both VQGAN image encoders while freezing the rest of the model. We then froze the VQGAN weights and allowed the base transformer to update (see Table 3). We attribute the 1.81% improvement in MAE, along with the improvements in FID and IS, to the finetuning of both the VQGANs, as it improved the consistency of image patch tokens. This provided the checkpoint with more reliable image patches to generate from. However, swapping the HPA VQGAN with an OpenCell one led to a similar losses of distribution information seen in Fig. S7. This could be because the model overfits before being able to learn probabilities across tokens. The learning obstacle comes from the possibility that images patches within the finetuned OpenCell VQGAN have sufficient (or even more) pixel consistency with the images, but the patch positional indices are misaligned with those of the HPA VQGAN. These findings are consistent with those found in analogous text-to-image works utilizing diffusion models. We did not find that finetuning improved the model's sequence reconstruction ability (see Table S9).

## 6 Discussion

### 6.1 CELL-E Comparison

In Table S10 and Table S11, we compare the performance of image localization prediction from scratch for CELL-E 2 and CELL-E. On the OpenCell validation set, CELL-E under-performs CELL-E 2 both before and after finetuning with regards to Nucleus Proportion MAPE. CELL-E 2 achieves worse Image and PDF MAE metrics before finetuning, however after finetuning CELL-E 2 achieves a 2.2% improvement for Image MAE and 1.7% for PDF MAE. On the contrary, CELL-E performs better with respect to image fidelity metrics SSIM and FID.

With respect to generation time, we found that the CELL-E 2 with hidden size of 480 was able to generate a prediction $65\times$ faster than the CELL-E model. This is a result of CELL-E 2's capability

to generate a prediction in a single step (.2784 seconds), which enables the advent of large-scale *in silico* mutagenesis studies.

## 6.2  *De novo* NLS Design

CELL-E 2's bidirectional integration of sequence and image information allows for an entirely novel image-based approach to *de novo* protein design. We applied CELL-E 2 to generate NLSs, which is a short amino acid sequence motif that can relocate a target protein into the cell nucleus when append to the target protein. In this case, our choice of the target protein is the Green Fluorescent Protein (GFP), a common protein engineering target [62–64] that is non-native to the human proteome and absent in the datasets. NLSs are usually identified by experimental mutagenesis studies or *in silico* screens that search for frequent sequences in nuclear proteins [51, 65]. However, these methods may yield candidates that are highly similar to known ones or not specific to the target protein. A more recent approach uses machine learning on sequence identity to augment featurization and statistical priors [17], but it is limited by the distribution of training samples due to the scarcity of experimentally verified NLSs. CELL-E 2 overcomes these limitations because it does not rely on explicit labels, and can therefore leverage significantly more unlabelled image data.

We generated a list of 255 novel NLS sequences for GFP using the procedure described in Appendix D.2. Briefly, we insert mask tokens of set length in a GFP sequence and task model with best sequence in-filling performance (`OpenCell_2560`) to fill in the masked amino acids, conditioned on a threshold image generated from the nucleus image (via Cellpose segmentation [66]). To verify the accuracy of the prediction, we pass the predicted sequence through the best performing image model (`HPA Finteuned (Finetuned HPA VQGAN)_480`), and quantify the proportion of signal intensity within the nucleus of the predicted threshold image (Fig. S9). The NLS sequences were then ranked based on sequence and embedding similarity with known NLSs (see Appendix D.2). The list of candidates can be found in Appendix D.3. We found several NLS candidates with high predicted signal in the nucleus, but which were fairly dissimilar from any protein found within NLSdb [65].

Classical NLSs are characterized by having regions of basic, positively charged amino acids arginine (R) and lysine (K) [67, 68], and are categorized as "monopartite" or "bipartite", either having a single cluster of basic amino acids or two clusters separated by a linker, respectively [69]. We observed a postive correlation between percentage of R and K residues in our predicted NLSs and sequence homology with known NLSs (Table S12). The number of clusters per sequence followed a similar trend, with sequences with relatively low sequence homology (Max ID $\% \leq 33$) having at most 2 clusters in 88% of predictions (Fig. S10). The remaining predictions, if correct, represent non-classical NLSs.

To further verify our predicted sequences, we passed the predicted NLS appended to GFP through Deeploc 2.0 [32], a leading sequence-to-class protein localization model, which predicted 89% of generated sequences were nuclear localizing and 91% contained a potential nuclear localizing signal.

Similar to CELL-E, we observed high attention weights on documented localization sequences correlated with positive protein signal within the threshold image (Fig. S11). For sequences with high predicted nucleus proportion intensities, we observed high activation across the entire sequence (novel NLS and GFP residues), with some NLS weights being an order of magnitude higher than others across the GFP sequences (Fig. 3). On the contrary, predicted sequences with comparatively less predicted intensity within the nucleus had low activation across the sequence, with little to none in the proposed NLS. We observed similar amounts of attention placed on the nucleus image patches, which largely corresponded to the location of the predicted threshold patches.

## 7  Conclusion & Future Work

In this paper, we have presented CELL-E 2, a novel bidirectional NAR model for protein design and engineering. CELL-E 2 can generate both image and sequence predictions, handle multimodal inputs and outputs, and run significantly faster than the state-of-the-art. By pre-training on a large HPA dataset and fine-tuning on OpenCell, CELL-E 2 can achieve competitive or superior performance on image and sequence reconstruction tasks. However, one limitation of CELL-E 2 is its output resolution, which is currently ($256 \times 256$). This resolution may not capture the fine details of microscopy images, which may limit applications in real-world use where Megapixel images are ac-

Figure 3: Attention weights associated with positive signal within the predicted image. Tokens with higher attention weight associated with background patches (low signal) are not highlighted. See Appendix D.3 for more information about the visualization process. We show 3 sequences with the highest (left column) and lowest (right column, not included in Table S13) predicted nucleus proportion intensity. The NLS+GFP sequences are shown with the designed NLS boxed in red.

quired. Increasing the output resolution of CELL-E 2 is one direction for future work. Furthermore, the sequence prediction struggles with the prediction of large stretches of amino acids as opposed to singular masked positions. Within this work, we encountered a trade-off between sequence prediction quality and prediction speed which may be overcome by reformulating the masking strategy. Similar findings were seen when compared with CELL-E, where we found accuracy measurements to improve with CELL-E 2 at the detriment of image quality metrics. The in-order prediction sequence we utilized in this paper may serve as a bottleneck for protein engineering applications despite the speed advantages gained from using a NAR architecture.

Another direction for future work is to incorporate structural information into the sequence embeddings. CELL-E 2 can generate novel NLS sequences with similar properties to GFP but low homology to existing sequences. However, the current sequence embeddings are based on a language model that may not capture all the structural features of the proteins. These features may affect the image appearance and vice versa.

We believe that CELL-E 2 is a promising model for protein design and engineering. We hope that our work will inspire more research on bidirectional NAR models for this domain and other domains that involve multimodal data.

## 8  Acknowledgments.

This research is supported in part by NIH grants R01GM131641 (BH) and R35-GM134922 (YSS). B.H. is a Chan Zuckerberg Biohub Investigator. A.A. contributed to visualizations and demos. E.K. was responsible for exploration and code.

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
