# OpenReview forum: "CELLE-2: Translating Proteins to Pictures and Back with a Bidirectional Text-to-Image Transformer"
_NeurIPS.cc/2023/Conference — NeurIPS 2023 poster_

### Official Review · Reviewer_DiCT · 2023-06-21

**Soundness:** 2 fair
**Presentation:** 3 good
**Contribution:** 2 fair
**Rating:** 5
**Confidence:** 3

**Summary:**

In this work, the authors present CellBERT-E, a transformer-based model to generate protein localization images. The model takes in the nucleus and threshold images as well as amino acid (AA) sequences as input. The images are tokenized via VQGAN tokenizer and AA sequences are tokenized via pretrained protein language model. The model is pretrained via mask modeling for both AA and image tokens. Experiments show that CellBERT-E achieves reasonable results in protein localization prediction. The authors also include experiments concerning generating AA motifs from image input. Besides, CellBERT-E generates images in a non-autoregressive manner, which is more efficient than previous CELL-E model.

**Strengths:**

1. Application of the transformer-based generative model to protein localization prediction, a relatively new domain.
2. Leveraging non-autoregressive generation which is more efficient than previous work.
3. Application of the proposed model to generate short AA sequences from nucleus images. Such a setting is tested on NLS design which can be an interesting attempt.

**Weaknesses:**

1. Lack of baseline models. The authors only compare different settings of the proposed CellBERT-E but miss other baselines.
2. Experiments may not support the conclusion of the proposed method. It can be hard to tell the effect of hyperparameters based on current experimental settings.
3. Evaluation metrics may not fully reflect the performance on the application. Authors apply common metrics for image/sequence generation to evaluate the protein localization prediction performance. Domain details may be missed when only applying these metrics.

Please see "Questions" for more details.

**Questions:**

Major questions:
1. Major results in Table 1&2 only compare different settings of proposed CellBERT-E and miss the performance of other baseline models. Though this work focuses on a relatively new application that few works have investigated. The authors can at least include the comparison to CELL-E which the proposed method is based on.
2. The choice of hyperparameters needs more validation. In Table 1&2, the deeper the model is, the smaller the hidden size is. Why do the authors choose such settings in architecture exploration? Besides, it's hard to conclude the effect of hidden size and depth on the performance as they are changing together.
3. Table 1 reports several image metrics to show the performance of CellBERT-E. Are there other domain metrics that are important in protein localization prediction? Will these image metrics fail to capture such important details?
4. The authors mention the potential rapid overfitting on OpenCell containing limited data. Have the authors considered any data augmentation strategies to enlarge the training set?
5. In Table 2, all models achieve high cosine similarities. Is this because only 15% of AA sequence is masked? The authors can add the results of random sampling to better validate the effectiveness of proposed model.
6. In Table 3, models that perform well on FID are likely to perform poorly on the other 5 metrics. What could be the reason?
7. In Section 6.2, the authors use the predicted signal in nucleus from the image model to validate the generated NLS. I find it unconvincing since the image model trained on limited data can fail when new data is fed in.

Minor questions:
1. Typo in line 88, I suppose it should be "allowing images to be synthesized in relatively few steps".


**Limitations:**

Besides the structural information of proteins mentioned by the author. The limited training data can be another limitation of the work. Even HPA, the larger dataset in this work, contains only ~17,000 data which is noisy. The authors bring up this in Introduction. However, how the proposed method addresses the concern should be further clarified.

---

> ### Author Rebuttal · Authors · 2023-08-10
>
> Thank you for the feedback, we will address the concerns below:
>
> > ... The authors can at least include the comparison to CELL-E which the proposed method is based on.
>
> As the reviewer agrees, this is an incredibly new field of study. As such, there are no proper baseline models with which to compare. For image prediction, the only comparison would be to CELL-E. Our evaluations of different training data (CELL-E uses OpenCell only whereas CellBERT-E combines HPA with OpenCell) and language models (CELL-E uses the BERT model from [1] whereas CellBERT-E uses ESM-2) are implicitly comparing these two models and showing the improvements of CellBERT-E. However, we will include an explicit comparison with CELL-E in the final revision.
>
> > The choice of hyperparameters needs more validation. In Table 1&2, the deeper the model is, the smaller the hidden size is. Why do the authors choose such settings in architecture exploration? Besides, it's hard to conclude the effect of hidden size and depth on the performance as they are changing together.
>
> Hidden size in predetermined by the ESM-2 language model that we use as a backbone. Depth is based on the findings of the CELL-E paper, showing a correlation between depth and token prediction ability. We maximized the depth that based on the available VRAM in training. We will be including this rationale in Section B.2.
>
> > ...Are there other domain metrics that are important in protein localization prediction? Will these image metrics fail to capture such important details?
>
> “Nucleus Proportion MAPE” is included as a domain-specific metric which calculates the proportion of intensity within the cell nucleus. This is the most relevant metric to the problem. We include other image metrics for the sake of thoroughness and evaluation for future models.
>
> > The authors mention the potential rapid overfitting on OpenCell containing limited data. Have the authors considered any data augmentation strategies to enlarge the training set?
>
> We already utilize standard data augmentation techniques which involved randomly cropping the amino acid sequence (A.3) and randomly cropping and flipping on the image (A.4). HPA much larger and more diverse than OpenCell, and pretraining helps our model learn generalizable features for localization prediction.
>
> > In Table 2, all models achieve high cosine similarities. Is this because only 15% of AA sequence is masked? The authors can add the results of random sampling to better validate the effectiveness of proposed model.
>
> Masking 15% of the amino acids is a standard in unsupervised protein sequence learning. Could you clarify what “random sampling” means in this context, as the chosen positions to mask in Table 2 are selected at random.
>
> > In Table 3, models that perform well on FID are likely to perform poorly on the other 5 metrics. What could be the reason?
>
> FID measures the similarity between the synthesized images and the ground truth images in terms of the feature space of Inception v3. Models that performed well on FID have only seen one type of image during training (Table 3), and they may have learned to generate images that resemble that type. This may result in high FID scores, but low scores on the other metrics, which are more sensitive to the correct localization of the protein within the cell. FID is also not a stable metric, and can be affected by factors such as JPEG quality [2].
>
> > In Section 6.2, the authors use the predicted signal in nucleus from the image model to validate the generated NLS. I find it unconvincing since the image model trained on limited data can fail when new data is fed in.
>
> Our goal was to create a list of candidate NLS sequences for future experimental validation. The image prediction is used to filter out unlikely candidates and not for validation. It is part of the process to generate the NLS candidate list. A two-step process, which combines the reverse model of sequence infilling and forward model of image prediction, should increase the success rate.
>
> Our current validation of the generated NLS candidates is based on the biological knowledge about NLS, such as the enrichment of certain types of amino acids and sequence features (domain knowledge not provided to the model). As shown in Table S9 and Figure S10, our generated NLS candidates have a strong potential to be functional.
>
> To address the reviewer’s concern, we have additionally passed our generated NLS candidate sequences (appended to GFP) to DeepLoc 2.0 [3], an independent model predicting multiclass annotations of protein localizations. DeepLoc 2.0 predicted **89%** of the generated NLS candidates with nuclear localization and **91%** having potential nuclear localizing signals, clearly validating our model.
>
> > Typo in line 88, I suppose it should be "allowing images to be synthesized in relatively few steps".
>
> Thank you, we have updated the manuscript.
>
> > Besides the structural information of proteins mentioned by the author. The limited training data can be another limitation of the work. Even HPA, the larger dataset in this work, contains only ~17,000 data which is noisy. The authors bring up this in Introduction. However, how the proposed method addresses the concern should be further clarified.
>
> We agree that the limited and noisy training data is a challenge. We utilize a frozen pre-trained language model, ESM-2, which was trained on millions of protein sequences, threreby providing a rich source of prior knowledge from a large corpus. We have updated Section 4.2 to explain this rationale in more detail.
>
> [1] Rao, et. al. Evaluating Protein Transfer Learning with TAPE. Advances in Neural Information Processing Systems 2019
>
> [2] Parmar, et. al. On Aliased Resizing and Surprising Subtleties in GAN Evaluation. CVPR, 2022
>
> [3], Vineet, et. al. DeepLoc 2.0: multi-label subcellular localization prediction using protein language models. Nucleic acids research. 2022

---

> > ### Comment · Reviewer_DiCT · 2023-08-11
> >
> > I appreciate the authors' responses to my questions and adding extra experiments. The rebuttal has addressed most of my concerns and I have raised my score.
> >
> > In terms of question 5, what I meant is that all models achieve high cosine similarity in in-filling with 15% masking ratio. It would be interesting to see what's the cosine similarity if random sampling is applied for in-filling. This may help set another baseline and better evaluate the performance of proposed method.

---

> > > ### Author Response · Authors · 2023-08-13
> > >
> > > Appreciate the feedback. Thank you clarifying, we agree that a random sampling would make for a useful comparison and willl include it in the final revision.

---

### Official Review · Reviewer_pNFc · 2023-06-24

**Soundness:** 2 fair
**Presentation:** 3 good
**Contribution:** 3 good
**Rating:** 5
**Confidence:** 3

**Summary:**

This work proposes an image-sequence multimodal encoder to model the interdependencies between cellular image and protein sequence. The pre-trained ESM-2 protein language model is employed to extract protein sequence embeddings, and the pre-trained VQGAN is used to extract cellular image patch embeddings. A Transformer encoder is trained upon these two kinds of embeddings to model the interaction between image patches and amino acid residues. The Validation set performance of image prediction and sequence infilling are analyzed to demonstrate model design choices. The application on de novo NLS design shows the effectiveness of the proposed model.

**Strengths:**

+ The proposed multimodality learning framework of cellular images and protein sequences is technically sound, and such a multimodality learning setting is novel to my best knowledge.
+ The results on de novo NLS design shows the model could helpful in real-world applications.


**Weaknesses:**

- Important downstream applications and baseline methods are not investigated in the experiment section.

- The evaluation protocol of image prediction and sequence infilling is not that standard from the machine learning perspective.


**Questions:**

1. The proposed method learns image-enhanced representations of protein sequences. Such representation could be superior over pure protein sequence representations learned by ESM-2. The subcellular localization prediction benchmarks proposed by DeepLoc [a] and DeepLoc 2.0 [b] could be a good test field for such a hypothesis, where the proposed CellBERT-E can compare with various protein language models.
2. In the image prediction and sequence infilling experiments, authors report the performance on validation set. However, from a standard machine learning perspective, the validation set should be used for model selection, and another hold out test set serves for evaluation. Authors are strongly encouraged to align such a standard.

[a] Almagro Armenteros, José Juan, et al. "DeepLoc: prediction of protein subcellular localization using deep learning." Bioinformatics 33.21 (2017): 3387-3395.

[b] Thumuluri, Vineet, et al. "DeepLoc 2.0: multi-label subcellular localization prediction using protein language models." Nucleic acids research 50.W1 (2022): W228-W234.


**Limitations:**

In the conclusion section, authors have not sufficiently discussed the limitations of their current method. They are encouraged to discuss potential limitations in terms of effectiveness, efficiency and the scope of applications.

---

> ### Author Rebuttal · Authors · 2023-08-10
>
> > Important downstream applications and baseline methods are not investigated in the experiment section.
>
> We included a highly tangible downstream application in the discussion. There, we demonstrated the generation of new NLS sequences using in-filling. In the protein engineering space, identifying candidate sequences is an important first step. Our demonstration showed the capability for CellBERT-E to generate new sequences driving the engineered protein to the desired subcellular localization. We have additionally passed our generated NLS candidate sequences (appended to GFP) to DeepLoc 2.0 [1], which is an independent model predicting multiclass annotations of protein localizations from their sequences. DeepLoc 2.0 predicted **89%** of the generated NLS candidates with nuclear localization and **91%** having potential nuclear localizing signals, clearly validating our model. We hope to experimentally validate these sequences.
>
> With respect to baseline methods, we note that this is a new field of study lacking baseline methods to compare against. For image prediction, the only available comparison is CELL-E. We will add an explicit, direct comparison in the revised paper. For sequence infilling, our model is the first of its kind with this capability. Therefore, we hope that our CellBERT-E itself will serve as the baseline for future developments in this field.
>
> > The evaluation protocol of image prediction and sequence infilling is not that standard from the machine learning perspective.
>
> We have tried our best to match the standard practices. However, the lack of established baselines and the absence of suitable image-based benchmarks complicates the having a comprehensive assessment. Therefore, we designed our evaluation protocol considering practical limitations and needs within the protein engineering domain.
>
> > ...The subcellular localization prediction benchmarks proposed by DeepLoc [a] and DeepLoc 2.0 [b] could be a good test field for such a hypothesis, where the proposed CellBERT-E can compare with various protein language models.
>
> CellBERT-E was not intended to be a potentially superior protein sequence representation. Instead, it was developed as a tool to translate between the amino acid sequence of a protein and its functional properties, akin in spirit to AlphaFold and other protein structure prediction models. Sequence embedding is just a part of CellBERT-E model.
>
> We thank the reviewer for bringing up the idea that CellBERT-E could lead to a better protein sequence representation that is “image-enhanced”. If so, the protein representation in CellBERT-E could indeed outperform purely sequence-trained ones for certain tasks, such as localization prediction in discrete categories like in DeepLoc [2] and DeepLoc 2.0 [1]. For this paper, we focus on the image prediction and sequence infilling capabilities of CellBERT-E, which has important downstream applications in protein engineering. We hope that future studies will elucidate the potential of CellBERT-E protein representation in other applications.
>
> Additionally, for benchmarking the image prediction of CellBERT-E, we previously considered comparing the indicated localizations in the predicted images to the multiclass predictions of DeepLoc 2.0 [2]. However, such a comparison requires conversion between two distinct modalities, i.e., annotate the protein localization in the predicted images using a separate image segmentation model. Therefore, we were afraid that this comparison could be inconclusive and would not serve as a proper benchmark. Still, if the reviewer feels that such comparison is necessary, we will include it in the revision.
>
> >  ... the validation set should be used for model selection, and another hold out test set serves for evaluation...
>
> We agree that a train-validation-test split is a standard practice in machine learning, but we argue that it is not feasible or necessary for our task. Protein datasets are very scarce and expensive to obtain, as they involve complex and costly wet-lab experiments. We use the HPA dataset, which is the largest available dataset of protein images, but still contains only 17,268 proteins. (Small compared to the 617M proteins used in training the sequence model ESM-2). Splitting the HPA dataset into three subsets would reduce the amount of data for training and validation, and it may not reflect the true performance of our model on unseen data.
>
> Moreover, we use another dataset, the OpenCell dataset, for finetuning and evaluation. The OpenCell dataset contains live imaging data of a specific human cell line, which is more realistic and consistent than the HPA dataset. The OpenCell dataset contains only ~1000 proteins. Splitting this dataset into three subsets would further limit the data availability and diversity for our task. Therefore, we follow the precedent of a recent work [3] that uses the same OpenCell dataset and adopts a two-way split. We believe that this is a reasonable and practical choice for our task, given the data constraints and challenges.
>
> > In the conclusion section, authors have not sufficiently discussed the limitations of their current method....
>
> In Section 7, we will be including a longer discussion on limitations, including real-world implications of the limited resolution and potential ways to address them in future works. We also include a discussion on the trade-off between prediction quality and speed in the sequence prediction stage.
>
> [1] Thumuluri, Vineet, et al. "DeepLoc 2.0: multi-label subcellular localization prediction using protein language models." Nucleic acids research. (2022):
>
> [2] Almagro Armenteros, José Juan, et al. "DeepLoc: prediction of protein subcellular localization using deep learning." Bioinformatics (2017):
>
> [3] Kobayashi, H., et al. Self-supervised deep learning encodes high-resolution features of protein subcellular localization. Nat Methods (2022).

---

> > ### Comment · Reviewer_pNFc · 2023-08-13
> > **Post-Rebuttal Comments**
> >
> > Thanks for the response, which addresses most of my concerns. I admit that the evaluation on protein engineering/function-prediction benchmarks is out of the scope of this paper, though this could still be very interesting as a future direction.
> >
> > Considering the contribution of this work on a new topic, I increase my rating to 5: borderline accept.

---

> > > ### Author Response · Authors · 2023-08-13
> > >
> > >
> > > Thank you, we agree it will be an interesting future direction. Appreciate the feedback!

---

### Official Review · Reviewer_9W6q · 2023-07-05

**Soundness:** 3 good
**Presentation:** 3 good
**Contribution:** 3 good
**Rating:** 7
**Confidence:** 4

**Summary:**

This paper proposes a novel bidirectional transformer named CellBERT-E to generate accurate protein localization image prediction from the amino acid sequences. To solve the ignorance of the integration of sequence and image information in existing methods, CellBERT-E adopts a BERT-like architecture so that the model can generate both image and sequence predictions in a non-autoregressive (NAR) paradigm at a fast speed. Therefore, the model allows for bidirectional prediction, making the model a possible candidate for de novo protein design. The model is trained by reconstructing the masked tokens in both the amino acid sequences and images in an unsupervised manner. Benefiting from the pretraining on a large HPA dataset and finetuning on the OpenCell dataset, CellBERT-E achieves competitive or superior performance compared with SOTA methods, which is shown by extensive experiment results.


**Strengths:**

Originality:
The paper proposes bidirectional transformer for text-to-image translation, and explores how this model could be used for protein design. Therefore, the paper uniquely contributes to the field by making fast and accurate predictions for protein sequence or image generation in the non-autoregressive manner.

Quality:
The paper carefully designs the experiments to support the idea and make clear visualizations.

Clarity:
The paper effectively communicates its ideas and findings with clarity. The paper is well-written, and the logic is coherent. The authors clearly illustrate the details in each section and make the ideas transparent to readers.

Significance:
The paper focuses on the bidirectional prediction of amino acid sequences and protein localization images with the advantage of faster prediction speed and possibly better protein de novo design than models of auto-regressive manner. And the proposed model does outperform baseline models. Therefore, this work is a promising model for protein design and engineering and could inspire research on bidirectional NAR models for this domain.

**Weaknesses:**

1. More background knowledge on biological terms mentioned in the paper is required (or explained more clearly), e.g., what are nucleus images. Besides, the protein images shown in the manuscript and supporting materials are nice, but it's not easy for a non-expert in biology to interpret something useful from the figures.

2. Although the authors' presentation is quite clear in general, the details of the finetuning task provided in the paper are not enough. Besides, it would be better if the authors can provide any intuition on why the task is designed in this way.

**Questions:**

1. What's the function of nucleus images over here? According to the authors'  explanation, the nucleus images are passed to the encoder but their tokens are not masked and thus not reconstructed during bidirectional prediction. Can the authors specify what's the role of nucleus images in this case (if there are any biological backgrounds related please clarify briefly)?

2. In Section 7, the sentence "By pre-training on a large 310 HPA dataset and fine-tuning on CELL-E, ..." should be changed by replacing CELL-E with OpenCell.

3. Although it is easier to parallel Transformer encoder-based models during inference, NAR Transformer decoders do exist [1, 2] and can significantly accelerate the decoding speed. Considering the recent success of GPT-based models, it would be great if the authors can discuss a little bit on whether the encoder-only CellBERT-E can be modified to a decoder-based one.

4. The goal of this paper is to train a model for sequence-to-image and image-to-sequence generation, so I'm wondering whether it's possible to train an encoder-decoder model (and cross attention might be more helpful in explaining how the amino acid sequences and threshold images are related), and maybe the authors can explain a little bit how they determine the model architecture.

5. With respect to the finetuning stage, I'm wondering what's the task here, is it the same as the training procedure described in Fig. 2? Why not mask all the tokens in the threshold figure and unmask the sequences (similar to the illustration in Fig. S3), which is closer to what the model is trained for: generate images depicting protein subcellular localization from the amino acid sequences? Probably, such a gap will make the model performs worse in the real application settings.

6. For the different finetuning strategies mentioned in section 5.3, I'm wondering whether the authors have tried finetuning the whole model simultaneously and how the performance is compared with other finetuned models.

[1] Gu, J., Bradbury, J., Xiong, C., Li, V. O., & Socher, R. Non-autoregressive neural machine translation. arXiv preprint arXiv:1711.02281.

[2] Huang, F., Tao, T., Zhou, H., Li, L., & Huang, M. On the learning of non-autoregressive transformers. In International Conference on Machine Learning (2022).

**Limitations:**

The authors have partially addressed the limitations of their work, though there is space for improvement (see Strengths, Weaknesses, and Questions).

---

> ### Author Rebuttal · Authors · 2023-08-10
>
> >More background knowledge on biological terms mentioned in the paper is required (or explained more clearly), e.g., what are nucleus images.
>
> Thank you for the feedback, we have updated the language in the introduction and methods to more to clearly explain those terms.
>
> > Although the authors' presentation is quite clear in general, the details of the finetuning task provided in the paper are not enough.
>
> We adopt a selective finetuning strategy that involves different parts of the model (e.g. the image representation via the VQGAN encoders) at different stages. This is motivated by the idea of domain adaptation, which aims to improve the performance of a model on a new domain by leveraging the knowledge learned from a different domain. We will update the manuscript to make this more clear.
>
> >Besides, it would be better if the authors can provide any intuition on why the task is designed in this way.
>
> We design the task to parallel the cell-imaging workflow, while also offering additional insights. In experiments, biologists stain the protein of interest as well as nucleus as a spatial reference. Therefore, we use the nucleus as a fixed input. Multiple acquisitions are often needed over time to understand protein dynamics. We demonstrate how our model can generate images quickly *in silico*, thus increasing the potential throughput and efficiency of such studies.
>
> > Can the authors specify what's the role of nucleus images in this case (if there are any biological backgrounds related please clarify briefly)?
>
> The nucleus image serves as a conditional image with which the model makes a prediction of protein localization with respect to. A predicted localization image holds little informational value without a spatial reference with which to compare to. Nucleus images are obtained for reference in wet-lab imaging workflows as standard practice. We have updated the manuscript to include this information in Section 4.2.
>
> >In Section 7, the sentence "By pre-training on a large 310 HPA dataset and fine-tuning on CELL-E, ..." should be changed by replacing CELL-E with OpenCell.
>
> Thank you, this will be changed.
>
> > Although it is easier to parallel Transformer encoder-based models during inference, NAR Transformer decoders do exist [1, 2] and can significantly accelerate the decoding speed. Considering the recent success of GPT-based models, it would be great if the authors can discuss a little bit on whether the encoder-only CellBERT-E can be modified to a decoder-based one. ...
>
> We have not tried the decoder-based model, but it is an interesting research direction. We use an encoder-only model to leverage the parallelism and independence of the patches, which are crucial for protein image synthesis. A decoder-based model would introduce dependencies and sequentiality among the patches, which may affect the heatmap quality. We are not aware of any work that uses NAR Transformer decoders for image generation.
>
> > ..I'm wondering whether it's possible to train an encoder-decoder model ... and maybe the authors can explain a little bit how they determine the model architecture.
>
> Our goal is to enable in silico screening of protein candidates, which requires a fast and scalable model (enabled by an encoder-based NAR model [1]) that can generate high-quality protein images from sequences and vice versa. An encoder-decoder model with cross-attention would introduce additional computational complexity and latency, which may hinder the practical application of our method. It is an interesting path to explore though.
>
> > With respect to the finetuning stage, I'm wondering what's the task here, is it the same as the training procedure described in Fig. 2?
>
> We pre-train our model on HPA data, which is diverse and heterogeneous. We finetune it on OpenCell data, which is more realistic and consistent for protein image synthesis, but cointains a specific human cell line. This allows the model to adpt to the domain.
>
> > Why not mask all the tokens in the threshold figure and unmask the sequences (similar to the illustration in Fig. S3), which is closer to what the model is trained for: generate images depicting protein subcellular localization from the amino acid sequences?...
>
> We do not mask all the tokens in the threshold image and unmask the sequences, because we want our model to be able to generate both modalities from each other. Our goal is not only to produce protein images from sequences, but also to produce sequences from images. This is useful for applications such as protein annotation, identification, and analysis. Therefore, we mask both image and text tokens during finetuning, as we did during pre-training, to train a bidirectional model that can handle both sequence-to-image and image-to-sequence generation.
>
> > ... I'm wondering whether the authors have tried finetuning the whole model simultaneously and how the performance is compared with other finetuned models.
>
> We have not tried finetuning the whole model simultaneously, because we follow the common practice of using a fixed image codebook in text-to-image models [1-3]. A fixed image codebook allows us to leverage the pre-trained image features and reduce the computational cost of finetuning. We can train on consumer-grade hardware because freezing parts of the model requires less VRAM, which is limited if our model is fully finetuned.
>
> [1] Huiwen Chang, Han Zhang, Jarred Barber, A. J. Maschinot, Jose Lezama, Lu Jiang, Ming-Hsuan Yang, Kevin Murphy, William T. Freeman, Michael Rubinstein, Yuanzhen Li, and Dilip Krishnan. Muse: Text-To-Image Generation via Masked Generative Transformers, January 5 2023.
>
> [2] Ming Ding, Wendi Zheng, Wenyi Hong, and Jie Tang. CogView2: Faster and Better Text-to-Image Generation via Hierarchical Transformers, May 2022.
>
> [3] Oran Gafni, Adam Polyak, Oron Ashual, Shelly Sheynin, Devi Parikh, and Yaniv Taigman. Make-A-Scene: Scene-Based Text-to-Image Generation with Human Priors, March 2022.

---

> > ### Comment · Reviewer_9W6q · 2023-08-10
> >
> > Thanks for the author's response to my questions and concerns. I really appreciate that you have helped me better understand your contributions.
> >
> > The responses have almost resolved my concerns except for the last two points. Firstly, I think the finetuned model does not have to be generalizable because the finetuned model should be trained with domain-specific and task-specific knowledge. So I still think you can try to mask all the tokens in the threshold figure and unmask the sequences and I believe this will improve the model performance in the specific task. Secondly, I agree that finetuning the model partially is more efficient, but I think it's still helpful if the authors can compare the results with finetuning the whole model and see whether the finetuning strategy the authors adopt is comparable with simply finetuning the whole model.

---

> > > ### Author Response · Authors · 2023-08-13
> > >
> > > In this work, we sought to leverage more general models for specific tasks without needing to constrain them to a specific task.  We agree that masking all the tokens in the threshold and unmasking the sequence will improve the performance. In a more specific study this is absolutely the direction we will go in.
> > >
> > > To be more clear about finetuning, while we agree it may be an interesting direction, we are unable to hold the entirely unfixed model in our available hardware due to the increased memory required to contain the gradients.

---

> > > > ### Comment · Reviewer_9W6q · 2023-08-18
> > > >
> > > > Thanks for the further response from the authors. I really appreciate your understanding of my concerns and illustrating the possible direction to improve the work.

---

### Official Review · Reviewer_ikCN · 2023-07-06

**Soundness:** 4 excellent
**Presentation:** 4 excellent
**Contribution:** 4 excellent
**Rating:** 7
**Confidence:** 3

**Summary:**

The authors propose a new architecture, CellBERT-E, for producing flexible embeddings that encode combinations of protein amino acid sequences and protein localization images. It can be used to generate localization images given a sequence and vice versa. Compared to its predecessor, it has many favorable characteristics; it was trained on slightly more data, is faster, and is bidirectional. It performs well on benchmarks.

**Strengths:**

The paper is exceptionally well-written and the method clearly improves on its predecessor, CELL-E, in multiple ways. Evaluation is thorough and carefully analyzed. The architecture is well-suited to the task and thoughtfully constructed.

**Weaknesses:**

My only real criticism is that the subject matter is quite niche (even to the point where some of the significance of this is lost on me), and I suspect that it will not interest most NeurIPS readers per se. That being said, I think the three-way multimodal architecture is well-executed and potentially worth acceptance as a nice case study in its own right.

**Questions:**

- Why do you clip logit values instead of softmaxing them (Supplement section B.1)?
- Varying depth along with width in Table 1 is a little awkward — it would be good to control for the effect of depth vs width. How do the parameter counts of the different models compare?
- Why are the HPA FID scores so erratic? E.g. it seems like HPA_640 achieves good scores in almost all categories but severely underperforms the other models in terms of FID.
-     “We also visually inspected some of the generated protein images (Fig. S6, Fig. S7). The output images from the OpenCell models appeared realistic and consistent with the ground truth labels, but they had low entropy in the predicted distribution. This suggests that the models learned to assign high probability to correct tokens, but failed to capture the uncertainty and variability of other valid selections. This could be attributed to the rapid overfitting of the OpenCell models, which limited their generalization ability.”

    Do the datasets contain the kind of diversity you allude to here (e.g. multiple protein images per sequence)? Is there any reason to expect the models to learn wide distributions over tokens?
-     "Most models had low performance on this task in terms of reconstruction. This is understandable because the models learned to generate amino acids that were common or frequent in the dataset, but not necessarily correct for the specific sequence."

    This is surprising to me, especially in light of Table S4. Why would ESM + image be so much worse than ESM alone? What exactly was the experimental setup for Table S4.



**Limitations:**

The authors include a thorough discussion of limitations.

---

> ### Author Rebuttal · Authors · 2023-08-10
>
> >My only real criticism is that the subject matter is quite niche (even to the point where some of the significance of this is lost on me), and I suspect that it will not interest most NeurIPS readers per se. That being said, I think the three-way multimodal architecture is well-executed and potentially worth acceptance as a nice case study in its own right.
>
> We appreciate the recognition of our three-way multimodal architecture as a nice case study. We also acknowledge that the subject matter may seem niche to some NeurIPS readers, but we would like to emphasize its relevance and significance to the broader machine learning community.
>
> First, our work aligns with the NeurIPS goal of fostering submissions for “Machine learning for sciences (e.g. climate, health, life sciences, physics, social sciences)”, which we believe is an important and growing area of interest. Second, our work has a high potential impact on the field of biology, especially for studying biological pathways and protein engineering. By enabling the in silico replication of experiments, our method can drastically reduce the experimental time and cost, as well as increase the number of possible targets to screen. Third, our work showcases a novel application of multimodal learning that integrates different types of data in a principled way. We believe that this approach can inspire other researchers to explore similar problems that involve multiple modalities and complex relationships.
>
> > Why do you clip logit values instead of softmaxing them (Supplement section B.1)?
>
> We would like to clarify that we do apply softmax to the logit values in our model, as described in Supplement section B.1. The clipping operation that we mention in the same section is not applied to the logits, but to the pixel values in the output heatmap. This is a post-processing step that we use to handle out-of-range values that may occur from the latent space interpolation within the VQGAN. We apologize for any confusion that this may have caused, and we will make this more clear in the manuscript.
>
> >Varying depth along with width in Table 1 is a little awkward — it would be good to control for the effect of depth vs width. How do the parameter counts of the different models compare?
>
> CELL-E [1] demonstrated that transformer depth and predictive performance have a strong positive correlation. The embedding size is pre-determined by the embedding dimensions used in the ESM-2 language model. We will include the rationale in Section B.1.  To accommodate our available compute hardware, we maximized the depth which could be fit in memory during training. We will include also include a table of parameter counts in the supplemental.
>
> >Why are the HPA FID scores so erratic? E.g. it seems like HPA_640 achieves good scores in almost all categories but severely underperforms the other models in terms of FID.
>
> The HPA dataset contains significant diversity in terms of cell type, image resolution, and antibody staining, which may cause inconsistencies in the model learnings. We have reported FID because it is a standard measure in text-to-image studies for natural images, but further investigation is needed to understand the behavior of FID in response to data in this domain. There is already ongoing debate on whether FID and IS truly correlate with image quality, as FID is not a stable metric. Recent work [2] has show that changing the jpeg quality from 100 to 75 can cause a difference of up to 20 FID points.
>
> > Do the datasets contain the kind of diversity you allude to here (e.g. multiple protein images per sequence)? Is there any reason to expect the models to learn wide distributions over tokens?
>
> > ...
>
> >  This is surprising to me, especially in light of Table S4. Why would ESM + image be so much worse than ESM alone? What exactly was the experimental setup for Table S4.
>
> The training objective required the model to reconstruct both image and text tokens. However, the HPA dataset, which is the largest available protein image dataset, containing just 17,268 proteins. This number pales in comparison to the 617M used in training ESM. We expect that performance would improve if model loss were only calculated on the masked amino acids.
>
>
>
> [1] Emaad Khwaja, Yun S. Song, and Bo Huang. CELL-E: Biological Zero-Shot Text-to-Image Synthesis for Protein Localization Prediction, May 2022. Pages: 2022.05.27.493774 Section: New Results.
>
> [2] Gaurav Parmar, Richard Zhang, Jun-Yan Zhu; Proceedings of the IEEE/CVF Conference on Computer Vision and Pattern Recognition (CVPR), 2022, pp. 11410-11420

---

> > ### Comment · Reviewer_ikCN · 2023-08-12
> >
> > Fair enough. I'll go up to 7.

---

> > > ### Author Response · Authors · 2023-08-13
> > >
> > > Thank you.  We appreciate the feedback!

---

### Decision · Program_Chairs · 2023-09-21

**Decision:**

Accept (poster)

**Comment:**

All reviewers agree that this paper should be accepted. This paper borrows the recent work in text-to-image generation that tokenized the image information using vector quantized approach and performs the text-to-image generation task using non-autoregressive token generation, which is currently one of the two mainstream methods in image generation.
This paper formalizes the protein location problem as the text-to-image generation, i.e., from protein sequence to a Nucleus image, which is quite promising and works well in the experiments. The main concerns of this paper lie in:1) the background of this task being less introduced, making it hard to understand the literature. 2) The task is very new, and should include more baseline for comparisons.